# Seasonal Variations in Canopy Size and Yield of Rayong 9 Cassava Genotype under Rainfed and Irrigated Conditions

**Supattra Mahakosee [1], Sanun Jogloy [1,2,*], Nimitr Vorasoot [1], Piyada Theerakulpisut [3], Poramate Banterng [1], Thawan Kesmala [1], Corley Holbrook [4] and Craig Kvien [5]**

[1] Department of Agronomy, Faculty of Agriculture, Khon Kaen University, Khon Kaen 40002, Thailand
[2] Peanut and Jerusalem Artichoke Improvement for Functional Food Research Group, Department of Agronomy, Faculty of Agriculture, Khon Kaen University, Khon Kaen 40002, Thailand
[3] Department of Biology, Faculty of Science, Khon Kaen University, Khon Kaen 40002, Thailand
[4] USDA-ARS, Crop Genetics and Breeding Research Unit, Coastal Plain Experimental Station, Tifton, GA 31793, USA
[5] Crop & Soil Sciences, The University of Georgia, Tifton, GA 31793, USA
**\*** Correspondence: sanjogloy@gmail.com; Tel.: +66-43-202-209

**Abstract:** The objectives of this study were to investigate the effect of seasonal variation on canopy size, and the effect of canopy size on light penetration of 'Rayong 9' cassava under irrigated and rainfed conditions. Rayong 9 was planted under two water regimes in a randomized complete block design with four replications in May and November for two years. At final harvest, years were significantly different ($p \leq 0.05$) for biomass, shoot dry weight, and harvest index and contributed to large portions of total variations in shoot dry weight (56.8%) and HI (44.5%). Planting date was a significant source of variations in all measured characters, and it contributed to the largest portions of variations in biomass, storage root dry weight and storage root fresh weight (46.1–60.9%). Water regimes were not significantly different for most characters except for harvest index ($p \leq 0.01$). The canopy of the crop planted in May grew rapidly in early growth stages in the rainy season and then slowly after the rainy season. The canopy of the crop planted in November grew rapidly in the middle to the late growth stages. Irrigation did not significantly increase root yield although it slightly increased canopy development, leaf area index (LAI), light penetration and photosynthesis. Irrigation at the late growth stages of the crop planted in May significantly increased storage root yield. Irrigation at these growth stages helped maintain canopy development, LAI and light penetration.

**Keywords:** light penetration; leaf area index; photosynthesis; water stress; climatic factors

## 1. Introduction

Cassava (*Manihot esculenta* Crantz, $2n = 36$) is an important root crop that can be used for food, feed and fuels [1]. Cassava is a perennial shrub, and it is monoecious species. It is a cross-pollinated crop [2]. Although cassava can produce seeds effectively, the seeds are only used for crop breeding. Cassava production relies on clonal propagation of stem cuttings. The crop is cultivated in the tropical areas from ~30° N to ~30° S in northern Brazil, south central Africa, Micronesia, Polynesia, Indonesia, Laos, Vietnam, Cambodia and Thailand [3]. Thailand is the second largest producer of cassava after Nigeria [4]. Although cassava can be planted in all months of the year, the main planting dates in South East Asia are in the early rainy season (May−June) and the late rainy season (October−December) [5]. Planting date is the main factor causing a yield difference of cassava, due to the variations in climatic factors during crop growth [6]. In Thailand, the average cassava yield was estimated at 21.3 t ha$^{-1}$ [4],

and the yield gap was still quite large compared to the potential yield of 90 t ha$^{-1}$ [7,8]. Under optimum conditions, light intensity and area of light interception determine biomass accumulation [9–12]. Variation in light intensity is associated with seasonal variation and variation in altitude, while the area of light interception depends on canopy characters, such as canopy shape, leaf position, and branch orientation [13–16].

Canopy development is dependent on climatic conditions [17,18]. Temperature, relative humidity and solar radiation affect cassava growth and development, especially leaf emergence, leaf growth, leaf size and leaf longevity [19–22]. Growth and development of leaves are determined by LAI, which plays a major role in the light captured for photosynthesis, and leads to biomass accumulation and greater crop yield [23]. Although the crop yield is closely related to LAI [7,24–27], the highest yielding genotypes of cassava do not always have the highest LAI [28]. Leaf arrangement and leaf size can affect light penetration and light availability, especially in the lower parts of the canopy, and it increases light harvesting efficiency and affects leaf photosynthesis activity [29].

Although cassava adapts well to low fertility and drought, drought is still the main cause of yield reduction of cassava. Most cassava growing areas are under rainfed conditions, and they are at a high risk of drought, which causes severe yield reductions [1]. Drought delays canopy development, leaf size and the rate of leaf emergence and, therefore, it reduces photosynthesis and storage root yield of cassava [30–32]. Irrigation helps maintain the canopy size with long leaf size, leaf area and leaf area duration [28,30]. These parameters are important for photosynthesis and biomass accumulation. The yield increase in irrigated cassava is proportional to the amount of water applied to the crop [32], and, if water resources are available, irrigation is an excellent means to increase the yield of cassava.

The hypotheses underlying this study are that planting date and irrigation are the main causes of the difference in canopy size in cassava and contribute to yield differences. Canopy size is an important parameter determining light penetration and photosynthesis of cassava. Information on the effects of seasonal variation on canopy size in cassava under irrigated and rainfed conditions is limited. Thus, the objectives of this study were to investigate the effect of seasonal variation on canopy size and the effect of canopy size on light penetration of Rayong 9 cassava under irrigated and rainfed conditions. The information obtained from this study will be useful for water management of cassava in different planting dates to obtain higher canopy size and yield.

## 2. Materials and Methods

### 2.1. Plant Materials and Experimental Design

The cassava variety, Rayong 9, was used in this study. Rayong 9 was released by the department of agriculture in Thailand. It is a non-forking type and was selected as a representative of high yielding varieties for this study. This variety was selected because it has a high yield and high starch content. Its yield potential is similar to other newly-released varieties in Thailand [33]. The field experiment was conducted at the Field Crop Research Station of Khon Kaen University, Thailand (16°47 N and 102°81′ E, 195 m above sea level). Rayong 9 was planted under two water regimes (irrigated and rainfed conditions) in a randomized complete block design (RCBD) with four replications for two planting dates (May and November) and two years (2015 and 2016).

May and November are the main cassava planting months in Thailand [5]. Cassava planted in the early rainy season in May receives rainfall during the early growth stages and drought at late growth stages until harvest. Cassava planted in the late rainy season in November can use stored soil moisture for crop establishment and survive through the dry period until the rainy season begins in May.

### 2.2. Crop Management

A chisel plough was used for soil preparation at a depth of 30−60 cm to break the hardpan. Sunhemp (*Crotalaria juncea*) was planted as a green manure and cattle manure was applied at the rate of 6250 kg ha$^{-1}$, then spread onto the soil followed by conventional tillage to prepare uniform

soil for the experiment. Soil ridges were made at the spacing of 1 m between the ridges to prepare experiment plots. Stem cuttings used in the experiment were prepared separately to obtain the cutting at the same age of nine to ten months for both planting dates. Healthy and uniformity stems were selected and harvested, discarding the two ends of the stem. Stem cuttings of 20 cm with three to six buds were used for planting. The cuttings were immersed into water containing thiamethoxam (3-(2-chlorothiazol-5-ylmethyl)-5-methyl-(1,3,5)-oxadiazinan-4-ylidene-N-nitroamine 25% water dispersible granules) at the rate of 4 g 20 $L^{-1}$ water for 20 min and incubated in a gunny sack at ambient temperature for three days, and selected for uniform budding, for planting.

Cuttings were inserted into the soil on the soil ridges at the spacing of $1 \times 1$ m. Thirty-five cuttings were planted in the $5 \times 7$ m plot. For two water regimes of each planting date, there were eight plots (units) total. An overhead sprinkler irrigation system was installed in each plot to apply water to the crop. Tensiometers were installed in the first replication at 20 and 40 cm soil depth to monitor soil moisture. Water was applied when soil suction at 40 cm soil depth was lower than −30 kPa. Water for the irrigation treatment was applied from planting to harvest, and water for the rainfed treatment was applied for one month after planting to ensure uniform plant establishment. Water for the rainfed treatment was stopped one month after planting (MAP) until harvest.

Manual weeding was done at one and two months after planting. Fertilizer (N:P:K) was applied after weed control following recommendations by Howeler [34]. Mealybug (*Centrococcus insolitus*) and red mites (*Eutetranychus orientalist* (Klein) were serious insect pests in the experiment, and experimental field were a monitor for the pest incidence in the field. Pests were controlled by weakly application (for three weeks) of thiamethoxam (3–(2–chloro-thiazol–5–ylmethyl)–5–methyl–(1,3,5) –oxadiazinane–4–ylidene–N–nitroamine 25% WG (at the rate of 75 g $ha^{-1}$) for control of mealy bug, amitraz (N-methylbis (2,4-xylyimino methyl)ameine 20% w/v EC) at the rate of 375 mL $ha^{-1}$ and pyridaben (4-chloro-2-(1,1-dimethylethyl)-5-[[4-(1,1-dimethylethyl)phenyl]methyl]thio]-3(2H)-py-ridazinone) at the rate of 187 g $ha^{-1}$ for control of red mites.

### 2.3. Data Collection

#### 2.3.1. Weather Condition and Soil Properties

Weather data were recorded for minimum and maximum temperature, rainfall, relative humidity and solar radiation to a data logger (Watch Dog 2700 PCE Germany, Meschede, Germany). Soil samples at the depths of 0−30 and 30−60 cm were taken to analyze soil physical and chemical properties, including soil pH, organic matter, total nitrogen, available phosphorus, exchangeable potassium, electrical conductivity (EC), and cation exchange capacity (CEC).

#### 2.3.2. Canopy Size Parameters

Canopy height was measured as the lowest node of the stem or branch that had a green leaf to the highest green leaf of the plant from one plant in each plot at monthly interval starting at one MAP until harvest. Canopy width was measured at the same plant as the diameter of the canopy at a monthly interval from one MAP until harvest. Canopy area and canopy volume were calculated from these parameters.

#### 2.3.3. Leaf Area Index (LAI)

LAI was measured (non-destructive) by using a plant canopy analyzer (LAI 2000; LICOR, Lincoln, NE, USA) following Malone et al. [35]. LAI was measured at three locations in each plot at monthly intervals, starting at one MAP until harvest.

#### 2.3.4. Light Penetration

Light penetration was measured under the plant canopy and between rows at three locations in each plot (11:30 am to 13:00 pm) at monthly intervals beginning one MAP until harvest by using line

quantum sensor (Licor 191) and data were recorded using a LI-1500 Data Logger (LI-Cor, Lincoln, NE, USA). Data were averaged to provide a single value for each plot.

### 2.3.5. Leaf Photosynthesis

Leaf photosynthesis was measured on one leaf in each canopy layer (upper, middle and lower layer) on one plant of each plot in each replication on the same plant at 3, 6, 9, and 12 months after planting. Canopy height was measured and divided into three equal layers. The 1st fully expanded leaf, the first leaf of middle and lower layers in the east direction were selected for measurement. The measurement was done on the sunny day during 08:00–12:00 am on the central lobe of the leaf by using leaf gas exchange portable photosynthesis (Licor 6400xt; LI-Cor, Lincoln, NE, USA). The conditions for photosynthesis were included light intensity (1500 $\mu$mol m$^{-2}$ s$^{-1}$), $CO_2$ concentration (400 $\mu$mol mol$^{-1}$) and block temperature (30 °C) in order to decrease confound of diurnal effects on photosynthesis. The data would be indicated of leaf photosynthetic performance under rainfed and irrigated conditions.

### 2.3.6. Biomass, Storage Root Yield, Starch Content and Harvest Index

At final harvest (12 MAP) eighteen plants from the harvest area in each plot were separated into leaf, petioles, stems and storage root. Total fresh weight of each plant part was recorded immediately, and a sample of more than 10% for each plant part was oven-dried at 80 °C for 72 h or until the weight was constant to determine dry weight. Storage root fresh weight was determined immediately after harvest, and a sample of 5 kg from each plot was used to measure starch content by specific gravity. A sample of approximately 10% of the storage root fresh weight was oven-dried at 80 °C for 72 h or until the weight was constant and used to calculate storage root dry weight. Biomass production was calculated based on shoot dry weight and storage root dry weight. Harvest index was computed as the ratio of total storage root dry weight to total biomass at the final harvest.

### 2.4. Statistical Analysis

The analysis of variance for individual planting dates was conducted, and error variances were tested for homogeneity. Combine analysis of two planting dates and two years with homogeneity were conducted following the procedure described by Gomez and Gomez [36]. Mean comparisons based on the least significant difference (LSD) were conducted for all planting dates by using MSTAT–C Version 1.42 program [37].

## 3. Results

### 3.1. Weather Condition

Maximum and minimum temperatures for cassava planted in May 2015 ranged from 16.4 to 43.9 °C, and from 8.9 to 29.5 °C, respectively (Figure 1a). Relative humidity ranged from 22.9 to 92.8%. The solar radiation from planting to harvest in May 2015 ranged from 6.4 to 25.4 MJ m$^{-2}$d$^{-1}$, and total solar radiation was 6,268.7 MJ m$^{-2}$ (Table 1). The total amount of rainfall was 883 mm. The true rainy season started from June and stopped in early October 2015; total rainfall was 727.5 mm. The dry period occurred in October to March (6−11 MAP) with total rainfall of 44.3 mm.

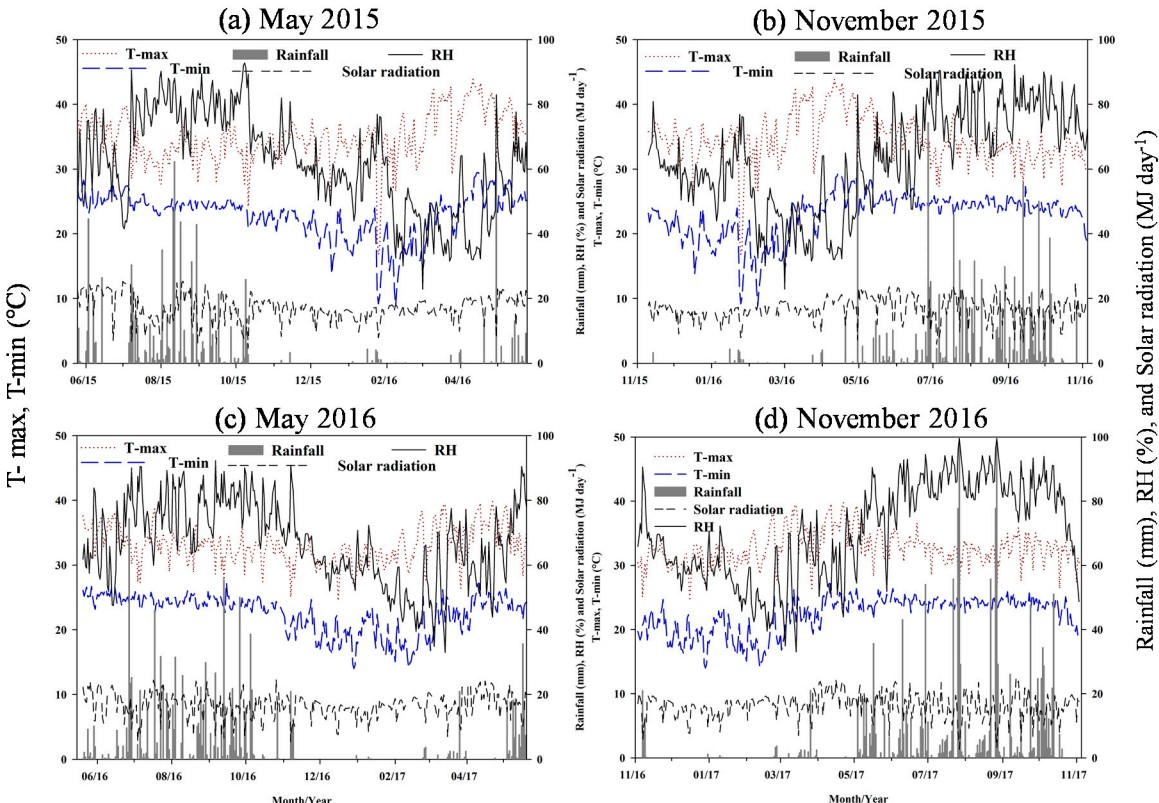

**Figure 1.** Maximum (···) and minimum (— —) temperature, rainfall (▉), relative humidity (—) and solar radiation (- - - -) at Khon Kaen University, planting dates May (**a**) and November (**b**) 2015 and May (**c**) and November (**d**) 2016.

**Table 1.** Monthly solar radiation (MJ m$^{-2}$) of Rayong 9 cassava under irrigated and rainfed conditions after planting in May and November 2015 and 2016. MAP, month after planting.

| MAP | | May 15 | May 16 | | Nov-15 | Nov-16 |
|---|---|---|---|---|---|---|
| 1 | May | 656.2 | 592.9 | Nov | 503.8 | 486.6 |
| 2 | Jun | 553.3 | 489.6 | Dec | 439.0 | 449.5 |
| 3 | Jul | 512.6 | 527.9 | Jan | 477.6 | 469.2 |
| 4 | Aug | 498.8 | 514.1 | Feb | 518.9 | 486.1 |
| 5 | Sep | 465.8 | 461.7 | Mar | 480.2 | 507.1 |
| 6 | Oct | 529.5 | 507.9 | Apr | 552.2 | 594.6 |
| 7 | Nov | 456.5 | 448.5 | May | 581.0 | 542.2 |
| 8 | Dec | 461.5 | 449.4 | Jun | 492.3 | 531.8 |
| 9 | Jan | 491.4 | 530.0 | Jul | 577.6 | 465.0 |
| 10 | Feb | 490.6 | 461.1 | Aug | 519.4 | 457.4 |
| 11 | Mar | 549.3 | 580.8 | Sep | 450.8 | 488.9 |
| 12 | Apr | 603.2 | 556.1 | Oct | 500.2 | 496.1 |
| Sum | | 6268.7 | 6120.0 | | 6093.0 | 5974.5 |

The maximum and minimum temperatures for cassava planted in November 2015 ranged from 16.8 to 43.9 °C, and from 8.9 to 29.5 °C, respectively (Figure 1b). Relative humidity ranged from 22.9 to 92.3%. Solar radiation ranged from 5.4 to 24.5 MJ m$^{-2}$ d$^{-1}$ with a total of 6093.0 MJ m$^{-2}$ (Table 1). The total amount of rainfall from planting to harvest was 1122 mm. The dry period was from planting to April (6 MAP) with a total rainfall of 111.8 mm. The rainy season was from May (7 MAP) to November (12 MAP) with a total rainfall of 1010.9 mm.

The maximum and minimum temperatures for cassava planted in May 2016 ranged from 24.5 to 39.8 °C, and from 14.0 to 27.3 °C, respectively (Figure 1c). Relative humidity ranged from 33.0 to 92.3%.

Solar radiation ranged from 5.4 to 24.5 MJ m$^{-2}$ d$^{-1}$ and the total was 6120.0 MJ m$^{-2}$ (Table 1). The total rainfall from planting to harvest was 1176 mm. The rainy season was from May to October (1–6 MAP) with a total rainfall of 981.2 mm. The dry period was from November to April (7–11 MAP) with a total rainfall of 44.2 mm. The end of crop duration had rainfall around 150.8 mm in May 2016.

The maximum and minimum temperatures for cassava planted in November 2016 ranged from 24.5 to 39.8 °C, and from 14.0 to 27.3 °C, respectively (Figure 1d). Relative humidity ranged from 33.0 to 99.6%. Solar radiation ranged from 7.7 to 24.8 MJ m$^{-2}$ d$^{-1}$ and the total was 5974.5 MJ m$^{-2}$ (Table 1). The total amount of rainfall from planting to harvest was 1469.3 mm. The dry period was from November to May (1–6 MAP) with a total rainfall of 44.2 mm. The rainy season was from June to November 2017 (7–12 MAP) with a total rainfall of 1382.5 mm.

Planting date May, the minimum temperature >25 °C and the maximum temperature <35 °C were 85 and 194 days (2015), 70 and 282 days (2016) respectively, whereas the crops planted in November were 88 and 272 days (2015), 39 and 300 days (2016) respectively (Table 2). However, the extreme temperatures (> 40 °C) occurred in 2015 (37 and 36 for planting date May and November respectively).

**Table 2.** Number of days with optimal temperatures, and suboptimal temperatures in May and November 2015 and 2016 planting date.

| MAP | | May 15 | | May 16 | | | Nov 15 | | Nov 16 | |
|---|---|---|---|---|---|---|---|---|---|---|
| | | T min > 25 °C | T max < 35 °C | T min > 25 °C | T max < 35 °C | | T min > 25 °C | T max < 35 °C | T min > 25 °C | T max < 35 °C |
| 1 | May | 19 | 7 | 25 | 8 | Nov | 0 | 16 | 0 | 30 |
| 2 | Jun | 17 | 15 | 10 | 24 | Dec | 0 | 20 | 0 | 30 |
| 3 | Jul | 2 | 26 | 12 | 29 | Jan | 0 | 22 | 0 | 30 |
| 4 | Aug | 4 | 25 | 12 | 28 | Feb | 0 | 18 | 1 | 24 |
| 5 | Sep | 3 | 25 | 3 | 29 | Mar | 22 | 2 | 13 | 8 |
| 6 | Oct | 0 | 15 | 0 | 30 | Apr | 24 | 6 | 8 | 18 |
| 7 | Nov | 0 | 19 | 0 | 30 | May | 24 | 6 | 8 | 18 |
| 8 | Dec | 0 | 23 | 0 | 30 | Jun | 15 | 19 | 6 | 25 |
| 9 | Jan | 0 | 21 | 0 | 29 | Jul | 11 | 26 | 1 | 30 |
| 10 | Feb | 1 | 11 | 3 | 15 | Aug | 13 | 28 | 2 | 30 |
| 11 | Mar | 14 | 4 | 7 | 15 | Sep | 6 | 29 | 9 | 25 |
| 12 | Apr | 26 | 4 | 9 | 15 | Oct | 1 | 37 | 1 | 35 |
| Sum | | 86 | 195 | 81 | 282 | | 116 | 229 | 49 | 303 |
| | | | 281 | | 363 | | | 245 | | 352 |

### 3.2. Soil Property and Soil Water Status

All experimented plots were planted in Yasothon soil series (Typic Paleustult). The soil textures were loamy sand and sandy loam (Table 3). Soil pH ranged from 7.01–7.41 except November 2015 planting date soil pH ranged from 6.27–6.50 at soil depths of 0–30 cm.

The soil of the four planting dates had organic matter at 0–30 cm soil depth ranging from 0.44–0.53% and total nitrogen ranging from 0.20–0.37%. The soils in all planting dates had available phosphorus ranging from 51.6–88.5 mg kg$^{-1}$. The critical level of available phosphorus in the soil is about 4–6 mg kg$^{-1}$ Bray II-extractable phosphorus [38]. Thus, the soil in this experiment had sufficient phosphorus for these crops. For soils used in all four planting dates, exchangeable potassium ranged from 30.8–54.6 mg kg$^{-1}$ at 0–30 cm soil depth. The soil in this experiment had insufficient exchangeable potassium. Exchangeable calcium ranged from 232.5–481.5 mg kg$^{-1}$ at 0–30 cm soil depth.

**Table 3.** Chemical and physical properties of the soil in the experimental fields at the depth 0−30 and 30−60 cm at planting date May and November 2015 and 2016.

| Planting Date | May, 15 | | Nov, 15 | | May, 16 | | Nov, 16 | |
|---|---|---|---|---|---|---|---|---|
| Depth (cm) | 0–30 | 30–60 | 0–30 | 30–60 | 0–30 | 30–60 | 0–30 | 30–60 |
| Soil Chemical | | | | | | | | |
| pH 1:1 $H_2O$ | 7.15 | 7.01 | 6.27 | 6.05 | 7.27 | 7.24 | 7.14 | 7.51 |
| Organic matter (%) | 0.44 | 0.38 | 0.46 | 0.32 | 0.5 | 0.44 | 0.53 | 0.18 |
| Total N (%) | 0.021 | 0.018 | 0.02 | 0.013 | 0.024 | 0.022 | 0.037 | 0.03 |
| Avail. P (mg kg$^{-1}$) | 61.2 | 56.5 | 51.6 | 42.1 | 77.6 | 76.1 | 88.5 | 27.9 |
| Exch. K (mg kg$^{-1}$) | 54.6 | 35.6 | 39.4 | 33.4 | 34.2 | 25.6 | 30.8 | 23.9 |
| Exch. Ca (mg kg$^{-1}$) | 338.8 | 386.8 | 232.5 | 277.5 | 359 | 364.5 | 481.5 | 460.5 |
| EC 1:5 $H_2O$ (ds m$^{-1}$) | 0.043 | 0.069 | 0.062 | 0.031 | 0.06 | 0.054 | 0.073 | 0.035 |
| CEC (c mol kg$^{-1}$) | 3.00 | 3.44 | 3.74 | 8.26 | 2.97 | 3.72 | 3.74 | 6.69 |
| Soil Physical | | | | | | | | |
| Texture Class | Loamy sand | Loamy sand | Sandy loam | Sandy loam | Sand | Sand | Loamy sand | Sandy loam |

EC, electrical conductivity; CEC, cation exchange capacity.

All of the crops under irrigation had soil water potential fell below the critical threshold (-30 KPa) for 13 and 9 days, in May 2015 and November 2015 planting date, respectively; and 1 and 0 days, in May 2015 and November 2015 planting date, respectively. Under the rainfed condition, planting date May 2015 and May 2016 had 182 days and 146 days, respectively (Table 4). However, in May 2016, the crops under rainfed subjected to low soil water potential in the early growth stage (1–3 MAP). The rainfed crops planted in November 2015 and 2016 were subjected to low soil water potential about 210 and 126 days respectively.

**Table 4.** The number of days on which the soil water potential fell below the critical threshold of Rayong 9 cassava under irrigated and rainfed conditions after planting in May and November 2015 and 2016.

| Planting Date | | 2015 | | 2016 | |
|---|---|---|---|---|---|
| | | Irrigated | Rainfed | Irrigated | Rainfed |
| | 1–3 MAP | 0 | 0 | 1 | 36 |
| | 4–6 MAP | 0 | 12 | 0 | 0 |
| May | 7–9 MAP | 0 | 90 | 0 | 20 |
| | 10–12 MAP | 13 | 80 | 0 | 90 |
| | **Sum** | **13** | **182** | **1** | **146** |
| | 1–3 MAP | 0 | 90 | 0 | 11 |
| | 4–6 MAP | 8 | 82 | 0 | 90 |
| Nov | 7–9 MAP | 1 | 38 | 0 | 25 |
| | 10–12 MAP | 0 | 0 | 0 | 0 |
| | **Sum** | **9** | **210** | **0** | **126** |

*3.3. Biomass, Storage Root Yield, Starch Content and Harvest Index*

At final harvest, years were significantly different ($p \leq 0.05$) for biomass, shoot dry weight, and harvest index, but they were not significantly different for storage root dry weight, storage root fresh weight and starch content (Table 5). Planting dates were significantly different ($p \leq 0.05$) for biomass, harvest index, storage root fresh weight, storage root dry weight, shoot dry weight ($p \leq 0.01$) and starch content ($p \leq 0.05$). The interactions between years and planting date were not significant for most characters except for shoot dry weight ($p \leq 0.01$). Water regimes were not significantly different for most characters except for harvest index ($p \leq 0.01$).

Table 5. Mean squares from combined analysis of Rayong 9 cassava under irrigated and rainfed conditions at planting dates May and November 2015 and 2016.

| Source | DF | Biomass | Storage Root Dry Weight | Shoot Dry Weight | Harvest Index | Storage Root Fresh Weight | Starch Content |
|---|---|---|---|---|---|---|---|
| Years (Y) | 1 | 372.0 **(22.7) [1] | 2.12$^{ns}$ (0.4) | 318.1 ** (56.8) | 0.074 ** (44.6) | 151.0$^{ns}$ (3.7) | 10.1$^{ns}$ (5.7) |
| Planting date (D) | 1 | 755.4 ** (46.1) | 351.9** (60.9) | 76.1 ** (13.6) | 0.019 ** (11.4) | 2437.8 ** (60.3) | 28.1 * (15.7) |
| Y *D | 1 | 61.7$^{ns}$ (3.8) | 0.2 $^{ns}$ (0.4) | 40.1 ** (7.2) | 0.000$^{ns}$ (0.0) | 3.9$^{ns}$ (0.1) | 15.1$^{ns}$ (8.5) |
| Error Y *D *R | 12 | 13.9 (10.2) | 9.6 (20.7) | 2.4 (5.1) | 0.002 (13.8) | 527.2 (13.0) | 5.3 (35.5) |
| Water regimes (W) | 1 | 0.7$^{ns}$ (0.0) | 0.2$^{ns}$ (0.0) | 1.6$^{ns}$ (0.3) | 0.014 ** (8.2) | 51.8$^{ns}$ (1.3) | 3.1$^{ns}$ (1.7) |
| Y*W | 1 | 159.7 ** (9.8) | 27.4 ** (4.7) | 54.7 ** (9.8) | 0.002$^{ns}$ (1.5) | 262.8 ** (6.5) | 3.1$^{ns}$ (1.7) |
| D *W | 1 | 32.2 * (2.0) | 45.9 ** (7.9) | 1.2$^{ns}$ (0.2) | 0.014 ** (8.7) | 351.8 ** (8.7) | 21.1* (11.8) |
| Y *D *W | 1 | 23. 2$^{ns}$ (1.4) | 0.1$^{ns}$ (0.0) | 20.8 ** (3.7) | 0.012 ** (7.2) | 8.1$^{ns}$ (0.2) | 0.1$^{ns}$ (0.1) |
| Error Y *D *R *W | 12 | 5.5 (4.2) | 2.39 (5.0) | 1.6 (3.4) | 0.001 (4.6) | 251.3 (6.2) | 2.9 (19.3) |
| Total | 31 | | | | | | |
| CV(Y *D *R) | | 12.6 | 18.7 | 12.5 | 7.5 | 14.6 | 8.5 |
| CV(Y *D *R *V) | | 7.9 | 9.2 | 9.9 | 4.3 | 10.1 | 6.2 |

$^{ns}$, *, ** Non-significant and significant at $p \leq 0.05$ and $p \leq 0.01$ probability levels, respectively. [1] Values in parentheses are percentages of sum squares.

The interactions between years and water regimes were significant for biomass, storage root dry weight, shoot dry weight and storage root fresh weight ($p \leq 0.01$), but they not significant for harvest index and starch content. Interactions between planting dates and water regimes were significant for all characters except shoot dry weight. Interactions between years, planting dates and water regimes were not significant for most characters except for shoot dry weight and harvest index ($p \leq 0.01$) (Table 5).

Years contributed to the largest portions of total variations in shoot dry weight (56.8%) and harvest index (45.2%), and moderate proportion for biomass (22.7%), but it had small contributions to storage root dry weight (0.4%), storage root fresh weight (3.7%) and starch content (5.7%). Planting dates contributed to the largest portions of total variations in biomass (46.1%), storage root dry weight (60.9%) and storage root fresh weight (60.3%), but it had moderate contributions to shoot dry weight (13.6%), harvest index (11.4%) and starch content (15.7%) (Table 5).

Results showed significant interactions between years and water regimes and planting dates × water regimes for most characters. The crops planted in November had higher biomass than those planted in May in both years. For planting in May 2015, the crop planted under irrigated had higher biomass than that under rainfed conditions, but the crop planted in May 2016 under irrigated conditions had lower biomass than did the crop planted under rainfed conditions. However, the crops started in November 2015 were not significantly different for biomass for both rainfed and irrigated conditions, but the crop started in November 2016 under rainfed conditions had higher biomass than did the irrigated crop. The crop planted in November 2015 had the highest biomass of 39.5 and 39.0 t ha$^{-1}$ for irrigated and rainfed crops, respectively (Figure 2a).

Storage root dry weight and storage root fresh weight showed similar patterns to biomass in both years. The crops planted in November in both years had higher storage root dry weight than did the crops planted in May. The highest storage root dry weight was found in the crop planted under rainfed conditions in November 2016 (21.6 t ha$^{-1}$) followed by the crop planted in November 2015 for both rainfed and irrigated conditions (20.9 and 20.4 t ha$^{-1}$, respectively). However, the crop planted under irrigated conditions in May 2015 had higher storage root dry weight than did the crop planted under rainfed conditions, but the crops planted in May 2016 under rainfed and irrigated conditions were not significantly different for storage root dry weight (Figure 2b).

The interactions between years and planting dates were significant for shoot dry weight, but had a small contribution (7.2%). Shoot dry weight after planting in November 2015 was higher than that after planting in November 2016 for both rainfed and irrigated crops. Moreover, the crops planted under irrigation in May were significantly different for shoot dry weight. The crop planted in May 2015 was higher than that in May 2016 for shoot dry weight, but the crops grown under rainfed conditions in May were not significantly different in both years (Figure 2c).

The crops planted in November had higher storage root fresh weight than did the crops planted in May in both years. The crop planted in November 2016 under rainfed condition had higher storage root fresh weight (56.9 t ha$^{-1}$) than did the crop planted under irrigated conditions (46.0 t ha$^{-1}$), but the crops planted in November 2015 under two water regimes were not significantly different (57.9 and 55.2 t ha$^{-1}$, irrigated and rainfed crops, respectively). However, the crop planted in May 2015 under irrigated conditions had higher storage root fresh weight (45.3 t ha$^{-1}$) than did the crop planted under rainfed conditions (31.4 t ha$^{-1}$), but the crop in May 2016 under both rainfed and irrigated conditions were not significant for storage root fresh weight (36.9 and 32.5 t ha$^{-1}$ for irrigated and rainfed crops, respectively) (Figure 2e).

Irrigated crops and rainfed crops were not significantly different for harvest index in most planting dates in both years, except for the crops planted in May 2016 in which the irrigated crop was significantly higher than the rainfed crop. In this experiment, harvest index values ranged from 0.52–0.68 for irrigated crops and 0.50–0.67 for rainfed crops (Figure 2d).

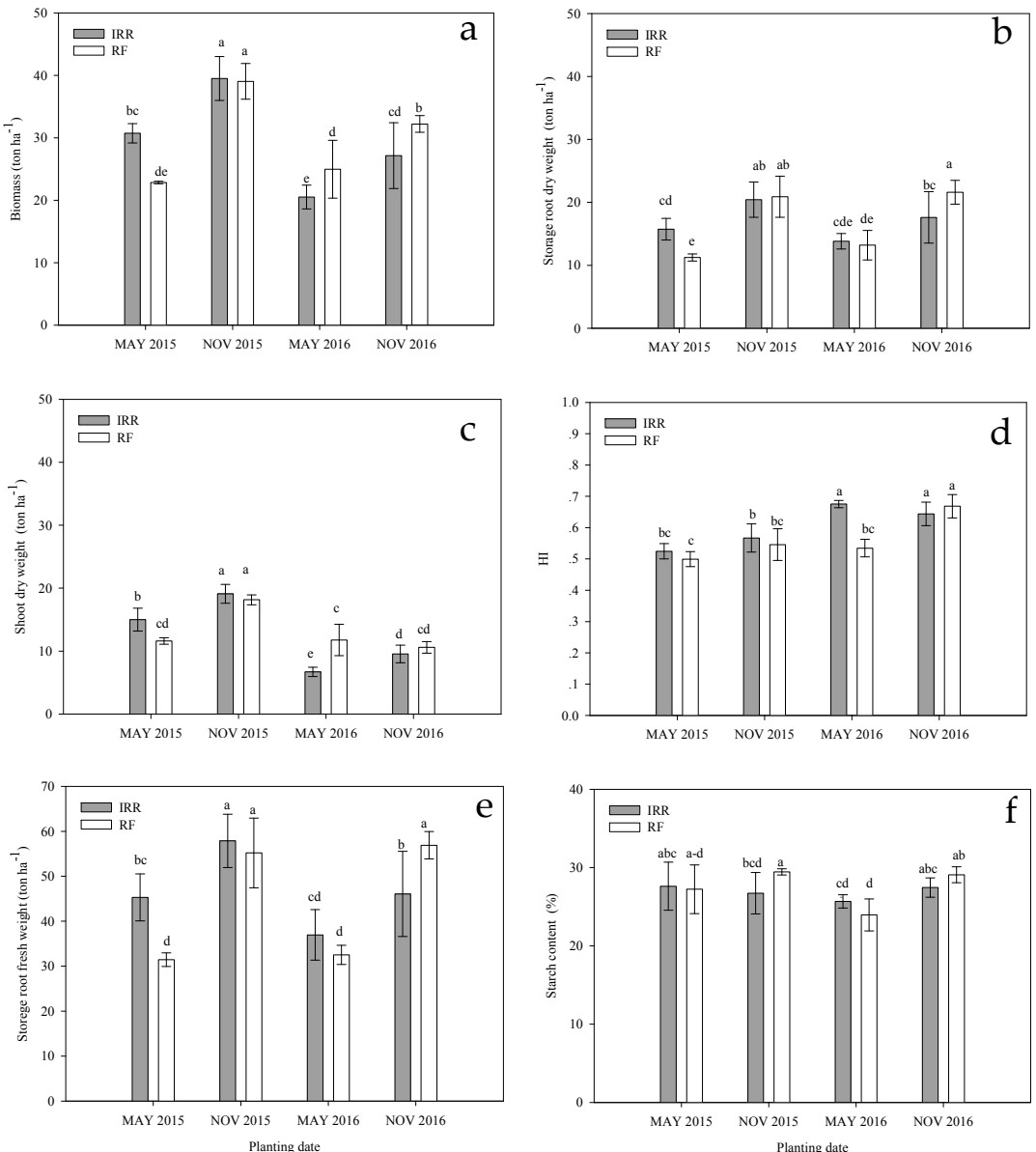

**Figure 2.** (**a**) Biomass; (**b**) storage root dry weight; (**c**) shoot dry weight; (**d**) harvest index (HI); (**e**) storage root fresh weight; and (**f**) starch content of Rayong 9 cassava genotype under irrigated and rainfed conditions at harvest (12 MAP), planting in May and November for two years. The bar is standard deviation.

Irrigated crops and rainfed crops were not significantly different for starch content in most planting dates except for the crops planted in November 2015. The crop grown under rainfed conditions had higher starch content than did the crop planted in November 2015 under irrigated conditions. In this experiment, starch contents ranged from 25.7–27.6 for irrigated crops and 23.9–29.5 for rainfed crops (Figure 2f).

### 3.4. Canopy Size

Traits related to canopy size consist of canopy height, which measures the height of green leaves, canopy width measuring the diameter of the canopy, canopy area measuring canopy coverage and canopy volume measuring three dimensions of the canopy. In order to obtain a better understanding on changes in the canopy from planting to harvest for rainfed and irrigated crops of two main planting dates, these parameters were evaluated at monthly intervals from one month after planting to harvest.

Canopy heights of rainfed and irrigated crops planted in May 2015 (Figure 3(1a)) increased with time and were highest in three months (for irrigated crop) and four MAP (for rainfed crop). The canopies declined from 200 cm to lower than 100 cm in December or seven MAP, and the reduction in canopy height was in agreement with the reduction in rainfall. After December, the rainfed crop had a higher reduction in canopy height than did the irrigated crop until harvest. It is interesting to note here that irrigation did not help maintain high canopy height because, in general, canopy heights of irrigated crops were similar to those of rainfed crops except at late stages of plant growth. On the other hand, for planting in May 2016 (Figure 3(2a)), the pattern of canopy height was similar to planting in May 2015, but plant canopy height for planting in May 2016 was smaller than those in May 2015. The peak of canopy height was 150 cm at five MAP for both water regimes, and the canopies were maintained at these heights until seven MAP. After seven MAP, canopy heights of both irrigated and rainfed crops declined and were lower than 100 cm. After nine MAP, canopy heights of both water regimes tended to increase until harvest.

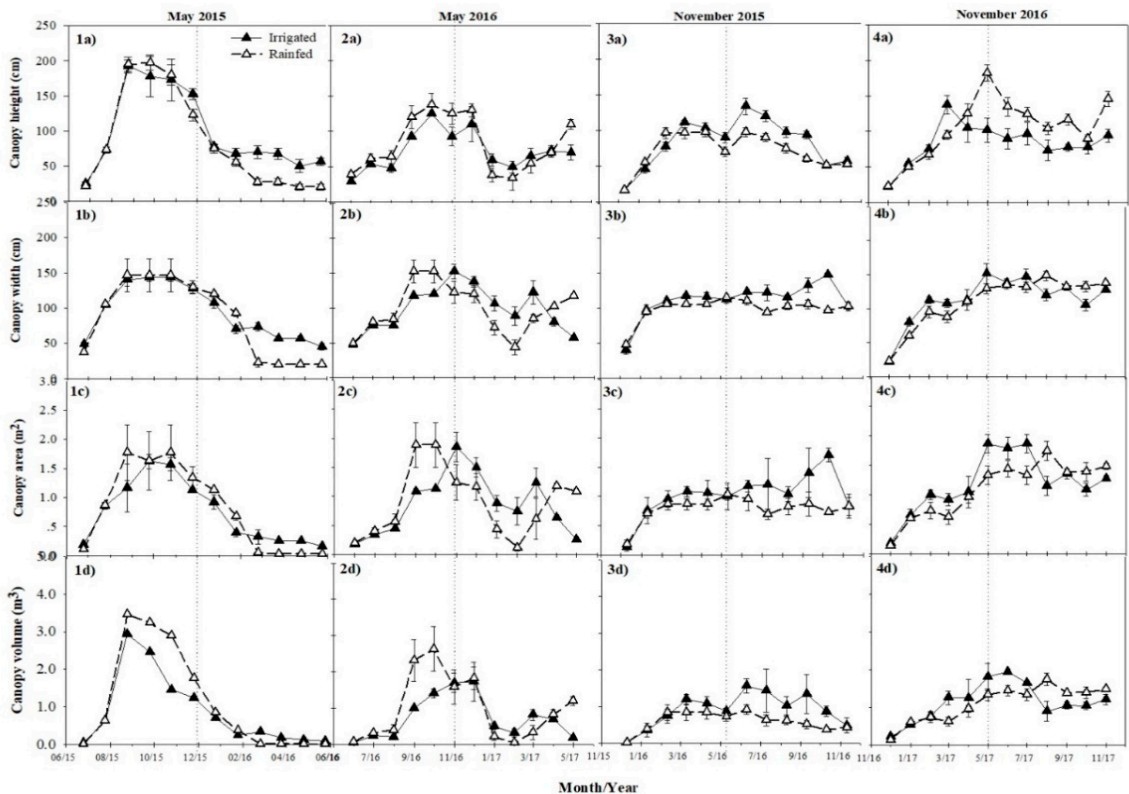

**Figure 3.** (**a**) Canopy height; (**b**) canopy width; (**c**) canopy area and (**d**) canopy volume of Rayong 9 cassava genotype under rainfed and irrigated conditions after planting in May and November for two years. The bar is standard deviation.

For planting in November 2015, both irrigated and rainfed crops increased canopy height with time until the peaks of their canopy heights at seven MAP (Figure 3(3a)). The highest canopy heights were recorded at 150 cm for the irrigated crop and 100 cm for the rainfed crop. After the peak, canopy heights for both irrigated and rainfed crops reduced, but the reduction occurred earlier for the rainfed crop. At late growing periods until harvest, both rainfed and irrigated crops were similar for canopy height. In this study, irrigating could increase canopy high only during the rainy season, but it did not affect canopy height in the dry season as the canopy heights were near 50 cm for both irrigated and rainfed crops. In contrast, canopy height for planting in November 2016 (Figure 3(4a)), the peak of canopy height was 180 cm (rainfed crop) at six MAP, and 140 cm (irrigated crop) at four MAP. However, both water regimes maintain canopy height at 100–150 cm until harvest.



Canopy widths of the irrigated and rainfed crops planted in May 2015 increased with time until they reached a peak at four MAP (Figure 3(1b)). The highest canopy widths were recorded at 150 cm for both rainfed and irrigated crops. After maintaining the peak for a month, canopy widths for both rainfed and irrigated crops declined sharply to about 50 cm for the irrigated crop and 25 cm for the rainfed crop at eight MAP (February). Canopy widths remained at these values until harvest. Similar to canopy height, irrigation generally did not affect the canopy width except at the late growth stages during the drought period. For the crops that were planted in May 2016 (Figure 3(2b)), canopy width peaked at 150 cm (at 4–5 MAP) for rainfed and 150 cm (at six MAP) for irrigated crops, then they declined at seven MAP to nine MAP. The rainfed crop was lowest at nine MAP (50 cm), and the irrigated crops were lowest at 12 MAP.

Canopy widths of the rainfed and irrigated crops planted in November 2015 (Figure 3(3b)) increased with time until they reached a peak at about 100 cm at three MAP, and the canopy widths were somewhat constant until harvest. Irrigated and rainfed crops were not significantly different for the canopy width, and had a similar pattern. The canopy width of the crop planted in November 2016 had a pattern similar to the crop planted in November 2015.

Canopy areas of the rainfed and irrigated crops planted in May 2015 increased from planting to four MAP when the crops had the highest canopy area of about 2.80 $m^2$ (Figure 3(1c)). Peak, canopy areas for both rainfed and irrigated crops, then sharply declined until they reached the lowest points at about 0.25 $m^2$ at eight MAP for the irrigated crop and near zero $m^2$ at nine MAP for the rainfed crop. Canopy areas of both irrigated and rainfed crops remained at these values until harvest. It was observed that irrigation affected canopy area only during drought periods, but the effect was still low. However, the canopy area of crops planted in May 2016 (Figure 3(2c)) had a pattern similar to the canopy width at the same time.

Canopy areas for rainfed and irrigated crops planted in November 2015 increased until they reached the highest values at seven MAP for the irrigated crop and 10 MAP for the rainfed crop (Figure 3(3c)), and then canopy areas remained somewhat constant until harvest. The highest canopy areas for both irrigated and rainfed crops were recorded at near 1 $m^2$ from three MAP until harvest. However, the rainfed crop had a lower canopy area than did the irrigated crop, yet the canopy area of the crop planted in November 2016 (Figure 3(4c)) had a pattern similar to canopy width during the same time period.

Canopy volumes for irrigated and rainfed crops planted in May 2015 increased from planting to a peak at four MAP, and, at the peak, the rainfed crop had a higher canopy volume than did the irrigated crop (Figure 3(1d)). The highest canopy volumes were 3 $m^3$ for irrigated and 3.5 $m^3$ for rainfed crops. After the peak, canopy volumes for both rainfed and irrigated crops sharply declined until they reached about 0.3 $m^3$ for the irrigated crop at eight MAP and near zero $m^3$ for the rainfed crop at nine MAP. The canopy volumes for both rainfed and irrigated crops maintained these values until harvest at 12 MAP. The canopy volume of the May 2016 planting, peaked at five MAP (2.8 $m^3$) for the rainfed crop and 2.0 $m^3$ for the irrigated crop (6 MAP). The canopy volume of both irrigated and rainfed crops declined sharply in January 2017 (6 MAP) (Figure 3(2d)).

Canopy volumes for irrigated and rainfed crops planted in November 2015 increased from planting to a peak at three MAP for rainfed crop and seven MAP for the irrigated crop (Figure 3(3d)), and the highest canopy volume was about 1.25 $m^3$ at the peak for the irrigated crop and about 1 $m^3$ for the rainfed crop. The irrigated crop had a higher canopy volume than the rainfed crop from four MAP until nine MAP, and they had similar canopy volumes from 10–12 MAP. The canopy volume of crops planted in November 2016, peaked at seven MAP (2.0 $m^3$) for irrigated and at nine MAP (1.8 $m^3$) for the rainfed crop (Figure 3(4d)).

*3.5. Leaf Area Index (LAI)*

LAI for irrigated and rainfed crops planted in May 2015 increased sharply and reached the highest values at four MAP for both irrigated and rainfed crops. The rainfed crop was slightly higher than the

irrigated crop (Figure 4a). The highest LAI were recorded at 5.26 and 5.01 for rainfed and irrigated crops, respectively. LAI for both rainfed and irrigated crops declined sharply after the peak, and the rainfed crop declined faster, resulting in a lower LAI than the irrigated crop from six MAP until harvest. It was clear that irrigation contributed to higher LAI in the irrigated crop during the drought period. However, for planting in May 2016, LAI for both irrigated and rainfed crops increased from planting to four MAP, then LAI were maintained at 3.54 to 3.69 (rainfed) and 2.57 to 3.03 (irrigated) until six MAP (Figure 4c). The highest LAI were 4.37 and 3.58 for rainfed and irrigated crops at seven MAP. The crops planted in rainfed conditions exhibited higher LAI than irrigated crops. LAI of both conditions declined sharply, after the peak and the rainfed crop declined faster. The pattern of LAI was similar to planting in May 2015.

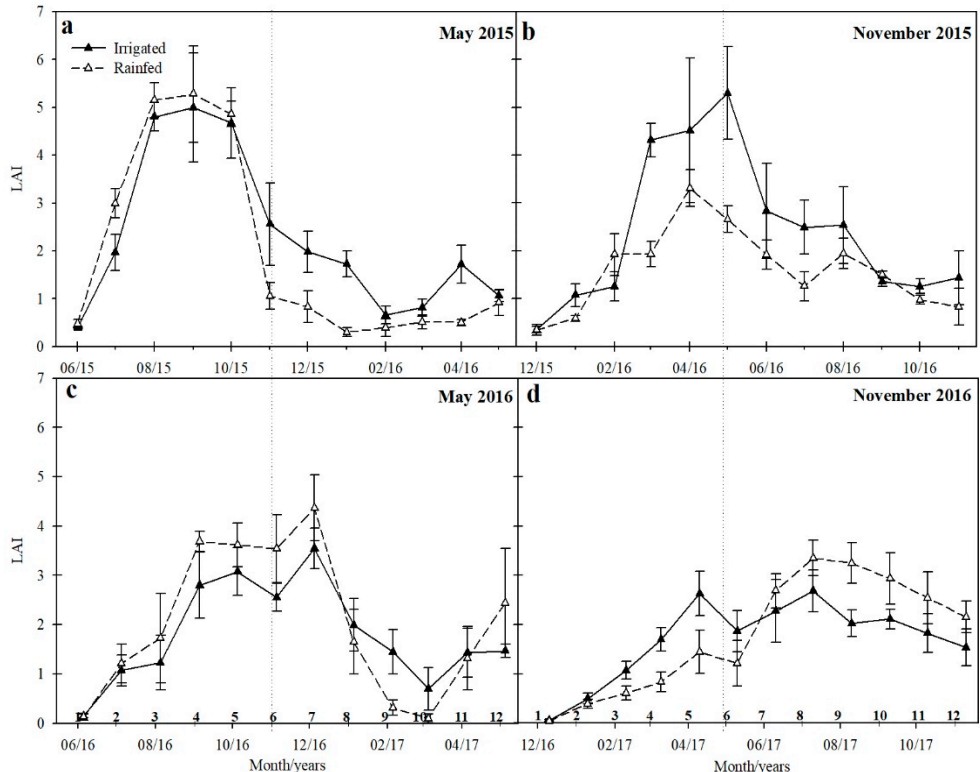

**Figure 4.** Leaf area index of Rayong 9 cassava under irrigated and rainfed conditions after planting in (**a**,**c**) May and (**b**,**d**) November 2015 and 2016. The bar is standard deviation.

LAI for the rainfed and irrigated crops planted in November 2015 and 2016 also increased at early growth stages until they reached maximum values of 3.31 at five MAP for the rainfed crop and 5.30 at six MAP for the irrigated crop planted in November 2015 (Figure 4b). In contrast, for crops planted in November 2016, LAI increased slowly at early growth stages until seven MAP for both irrigated and rainfed conditions. The highest LAI for the irrigated crop was 2.6 at five MAP with 2.6 LAI value, whereas the highest LAI for the rainfed crop was 5.8 at five MAP (Figure 4d).

*3.6. Light Penetration*

Light penetration through the canopy is negatively correlated to leaf area. Light penetrations of the irrigated and the rainfed crops planted in May and November, 2015 and 2016 were presented in Figure 5. In general, light penetrations declined from planting until it reached a low point and then it increases. The reduction in light penetration was due to the increase in leaf area and the increase in light penetration was due to defoliation as the crop was subjected to the dry period.

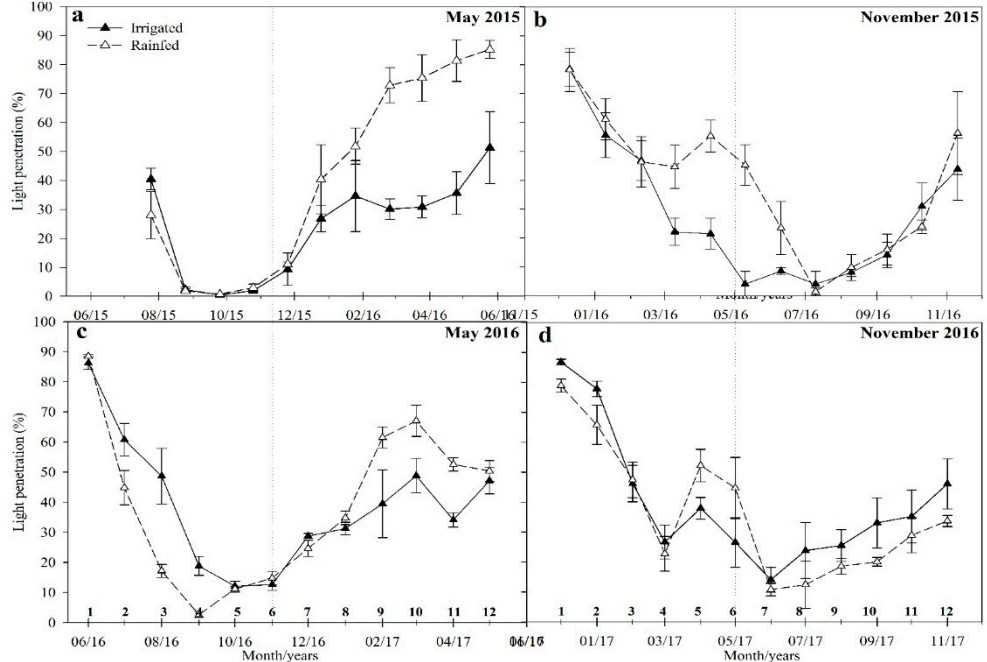

**Figure 5.** Light penetration of Rayong 9 cassava under irrigated and rainfed conditions after planting in (**a**,**c**) May and (**b**,**d**) November 2015 and 2016. The bar is standard deviation.

In general, rainfed and irrigated crops had similar patterns of light penetration both for the crop planted in May and the crop planted in November except for the dry periods. For the crops planted in May of both years, the light penetrations of rainfed crops had a sharper decrease than the irrigated crops until six MAP. The light penetration was near zero at 4–6 MAP (rainfed crop, May 2015) (Figure 5a) and four MAP (rainfed crop, May 2016) (Figure 5c). Light penetration then increased until harvest. Rainfed crops of both years had higher light penetration than irrigated crops of both years.

For the crops planted in November of both years, light penetration in both rainfed and irrigated conditions had slight decreases from planting to four MAP. The rainfed crop planted in November 2015 had higher light penetration than the irrigated crops at 4–7 MAP. Light penetration was near zero at eight MAP for both conditions. After that, light penetration increased until harvest (Figure 5b). The pattern of light penetration of crops planted in November 2016 was similar to the crops planted in November 2015. The lowest light penetration values ranged from 10% to 20% at seven MAP. The irrigated crop had high light penetration than rainfed crop. However, light penetration of both conditions trended to increase until harvest (Figure 5d).

*3.7. Photosynthesis*

Photosynthesis values from cassava leaves at the top, the middle and the lower parts of the canopy for the crops planted in November 2015, and May and November 2016 were presented in Figure 6. The crops planted in November 2015 had the highest photosynthesis at the top of the canopy for both irrigated and rainfed conditions at all sampling dates (3, 6, 9, and 12 MAP). Photosynthetic rates of the rainfed crop at the top of the canopy were higher than those of the irrigated crop at all sampling dates, whereas photosynthetic rates of the irrigated crop at the lower part of the canopy were higher than those of the rainfed crop. The highest photosynthesis rates were observed at nine MAP in both rainfed and irrigated crops. At this growth stage, the top leaf in the irrigated crop had a photosynthesis rate of 28.7 $\mu$mol m$^{-2}$ s$^{-1}$ and the top leaf in the rainfed crop had a photosynthetic rate of 30.9 $\mu$mol m$^{-2}$ s$^{-1}$.

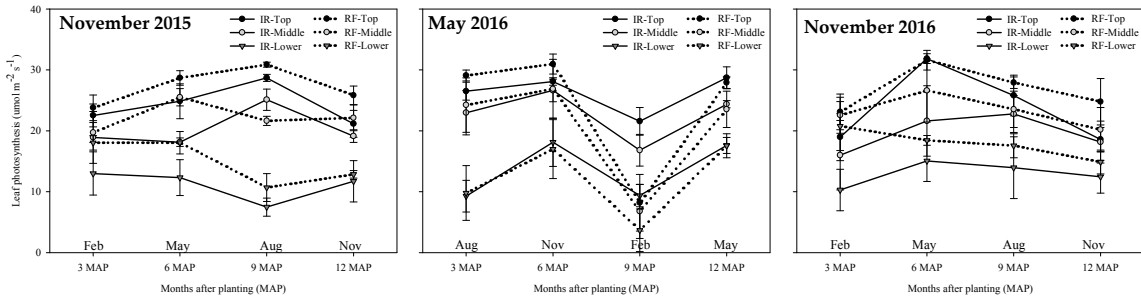

**Figure 6.** Photosynthesis of top, middle and lower cassava leaves in the canopy at 3, 6, 9, and 12 MAP under irrigated and rainfed conditions when planted in November 2015, and May and November 2016. The bar is standard deviation.

For the crops planted in May 2016, photosynthesis rates in the top of the canopy were highest for both rainfed and irrigated crops at three and six MAP. The rainfed crop generally had higher photosynthesis rates than did the irrigated crop. At six MAP, the photosynthesis rate in the top leaf of the irrigated crop was 28.6 µmol m$^{-2}$ s$^{-1}$, whereas, the photosynthesis rate in the top leaf of the rainfed crop was 31.0 µmol m$^{-2}$ s$^{-1}$. However, at nine MAP, the photosynthesis rates of cassava when grown under irrigation conditions (21.6, 16.8 and 9.4 µmol m$^{-2}$ s$^{-1}$ at top, middle and lower leaf, respectively) maintained a higher capacity than did cassava grown under rainfed conditions (8.2, 6.8 and 3.7 µmol m$^{-2}$ s$^{-1}$) for the top middle and lower leaf layers, respectively.

For the crops planted in November 2016, cassava maintained a high photosynthesis capacity at all growth stages. Photosynthetic rates evaluated at six MAP were highest among all growth stages. Photosynthetic rate of the top leaf in the irrigated crop was 31.9 µmol m$^{-2}$ s$^{-1}$, and the photosynthetic rate in the rainfed crop was 31.6 µmol m$^{-2}$ s$^{-1}$.

Across sampling dates, the increase in photosynthetic rates at the top of the canopy was associated with reductions in photosynthetic rates at the middle and the lower parts of the canopy and *vise versa*. Photosynthetic rates at the middle of the canopy were intermediate between the lower part and the top of the canopy, and there was a shift interaction between sampling time and water regime, meaning that sometimes the irrigated crop was higher than the rainfed crop, and vise versa.

## 4. Discussion

The study hypothesizes that planting date and water regime are the main causes of differences in canopy size and contribute to yield differences in cassava. In this study, planting in November resulted in higher biomass and root yield than did planting in May under both drought and well-irrigated conditions. The differences in biomass and yield for the two planting dates are not unexpected because the crops were grown under different climatic factors, such as temperature, relative humidity, solar radiation, and rainfall. These climatic factors are important for canopy growth and development [2,18,32], which ultimately becomes the source of light capture for the photosynthetic process, and subsequent assimilate partitioning and assimilate accumulation.

Planting in May 2015 and 2016 resulted in rapid growth of the canopy at the early growth stages, and the crops had the highest canopy size and the highest LAI at three and five MAP, respectively, whereas planting dates in November 2015 and 2016 produced the highest canopy size and the highest LAI at seven and six MAP, respectively. Differences in the peak times of growth and canopy development would be due to differences in climatic factors during the growing season, such as temperature, humidity, and solar radiation [20,21,28].

The cassava crops in this study were applied of fertilizer based on the recommended rate and sprayed with pesticides to control of major pests (Mealy bug and red mites). The canopy size of cassava crops planting in May 2015 was higher than those of the crops planting in May 2016, Although the meteorological data were similar, crops planted in May 2015 may have had a higher sum of solar

radiation (1722 MJ m$^{-2}$) during 1–3 months of canopy development stage, compared to the crops planting in May 2016 (1610 MJ m$^{-2}$) (Table 1).

Cassava planted under high solar radiation had higher growth, root density and yield than did the crop planted under shade as cassava required high solar radiation for photosynthesis and growth [22]. High temperature and high solar radiation during the growth period increased canopy size of cassava, including leaf size, the growth rate of new leaves, and LAI [7,19,22].

Cassava planted in May received higher solar radiation during May to August (1–3 MAP) and, therefore, it had a higher canopy growth than did cassava planted in November. Moreover, solar radiation during December to March was lower than solar radiation during May to August. Cassava planted in November also was under low temperature and low relative humidity during early growth phases. Temperatures lower than 17 °C and higher than 37 °C inhibit leaf development and reduce canopy growth of cassava [39,40]. However, the temperature is the main factor affecting photo-assimilate partitioning in cassava. The low temperature in the cool season reduced total biomass accumulation rates, but induced a higher partition of assimilates to storage roots than to shoots [41]. Low temperature also induced the growth of storage root earlier than high temperature [24]. The information obtained in this study, and in the earlier research, indicates that the crops planted in November had an earlier bulking of storage roots than the crop planted in May. According to Puangbut et al. [42], low temperature induced tuber development and partitioning of assimilates in Jerusalem artichoke, and the crop planted in the early rainy seasons had lower tuber dry weight than did the crop planted in the late rainy season, although biomass in the early rainy season was higher than in the late rainy season.

Although cassava planted in May had more rapid growth of canopy than did cassava planted in November, it had a lower root yield and biomass at harvest (Figure 2). At the storage root accumulation stage (6 to 10 MAP) [2], cassava planted in November had a more favorable temperature, solar radiation and relative humidity for leaf growth and development of most of the canopy than cassava planted in May. The results showed that canopy height, canopy width, canopy area and canopy volume of cassava planted in November were higher than those of cassava planted in May (Figure 3). In contrast, at similar stages, cassava planted in May was exposed to a lower temperature, lower solar radiation, lower relative humidity and lower soil moisture and, therefore, it had reduced canopy development, lower numbers of new leaves and lower photosynthesis. The results indicated that seasonal variations greatly affected leaf growth, LAI and canopy development of cassava. Similar results were also found in Australia (S.E. Queensland (latitude 27°37′ S) for example, LAI could be higher than 10.0 in mid-summer (January to March and April) for early planted crops and then declined to 0.1–0.3 in the cooler months (April to July) for all planting dates [20]. In cassava and other crop species, canopy size is closely correlated with LAI, light penetration, leaf photosynthesis and root yield. Yield is closely related to the area of light interception, light captures and conversion of light to biomass efficiency by photosynthesis [43]. In addition, the deeper light penetration in the canopy is important for photosynthesis of lower leaves. Investigation of light absorption and the vertical distribution of light in the tomato canopy was reported by Sarlikioti et al. [44]. The larger canopy of cassava in the May planting resulted in lower light penetration (> 20%) and shading of lower leaves. Shaded leaves in the lower canopy of cassava planted in May had lower photosynthetic rates than did shaded leaves of cassava planted in November at three MAP. Cassava planted in November had higher photosynthesis than did cassava planted in May because it received higher light intensity.

Temperature also affects leaf and canopy development and is an important factor affecting photosynthesis. A temperature range from 25 to 35 °C is optimum for maximum photosynthesis of cassava [45]. In this study, the crop planted in November maintained leaf photosynthesis in the top and middle parts of the canopy in all sampling dates, but the crop planted in May had lower leaf photosynthesis at nine MAP because it was subjected to low temperature and low relative humidity. El-Sharkawy [40] reported that the temperature below 20 °C decreased photosynthetic rate. Vongcharoen et al. [46] found that leaf photosynthesis was reduced in the cool season. According to

Alves [2], cassava increases root growth during the period of six to nine MAP, and this growth phase is critical for cassava yield, because cassava moves assimilates from leaves to storage roots [47,48]. Therefore, small canopy size, low leaf area and low photosynthesis resulted in low biomass and yield of cassava. In this study, the number of days for optimal temperatures trended to associate with the total biomass and storage root yield. However, the monthly sum of solar radiation was not associated with the total biomass and storage root yield. In addition, the monthly sum of solar radiation trended to associate with the canopy size (canopy height) of cassava (data not shown). Previous authors [2,47,48] raised the assumptions that irrigation water and planting date were important sources of variations in cassava yield. The results from this study agree with those assumptions. However, the interactions between planting date and water regime was also a significant source of variation in cassava yield. Large interactions may have been due to the uneven distribution of rainfall in 2015 and 2016. The responses to irrigation were different between irrigated crops and rainfed crops planted in May. The irrigated crop had higher root yield and biomass than did the rainfed crop in May 2015. In May 2016, the rainfed crop showed higher biomass and shoot dry weight than did the irrigated crop, but they were not significantly different for storage root yield. In May 2015, total rainfall in the rainy season was 727.5 mm, which was lower than the 981.2 mm for the crop planted in May 2016 and rain distribution of May 2016 was also longer than that of May 2015. The soil water potential lower than −30 KPa in crop duration of May 2015 (rainfed crops) had longer than May 2016 (rainfed crops). However, during 1–3 MAP the under rainfed crop in May 2016 was subjected to a water deficit for 36 days, while the rainfed crop in May 2015 were not subjected to water stress in early growth stage (Table 4).

When planting in May, irrigated and rainfed cassava were significantly different for canopy development at late growth stages. Although cassava planted in May was subjected to a cool and dry condition in the late growth stages, the crop under irrigation could maintain a larger canopy size until harvest, whereas the crop under rainfed condition lost leaves in the cool and dry season. The results were in agreement with those reported previously [29]. In addition, the irrigated crop had higher photosynthesis than did the rainfed crop at nine MAP because of higher relative humidity in the canopy. In contrast, the rainfed crop had lower relative humidity in the canopy, that induced stomatal closure and limited photosynthesis [46].

For the crops planted in November, the rainfed crops had insufficient water during early growth. Cassava, once it is established, can survive for several months without rain [49]. Cassava adapts to drought by deep rooting and partial closure stomata [27]. However, drought in early growth (1–5 MAP) reduced storage root yield of cassava more than 32% [29]. In this study, a drought at early growth stages did not have many detrimental effects on growth and yield of the rainfed crop planted in November. Although drought limited canopy development in rainfed crops at early growth stages, the crops could establish canopy more rapidly once it received rainfall in the early rainy season than the irrigated crops. In addition, the new leaves had higher photosynthetic rate [26]. Therefore, irrigated crops and rainfed crops planted in November were not significantly different for biomass production and cassava yield.

The results indicated that climatic factors, rainfall and irrigation were important for growth and development of cassava. The crops planted in May had a lower yield than did the crops planted in November because of lower temperature and a dry period in late growth stages that affected canopy development and photosynthetic accumulation. However, irrigation can alleviate the drought problem in the dry periods. Although the crops planted in November were subjected to low temperature and dry conditions in early growth, biomass and storage root yield were still higher than the crops planted in May. In addition, irrigated crops and rainfed crops were not significantly different for root yield because the crop under rainfed condition could survive during the drought period through stored soil moisture in the early growth stages. Once it received rain in the rainy season, the crop could resume growth and canopy development.

## 5. Conclusions

Cassava crops planted in May under rainfed and irrigated conditions were significantly different for storage root yield, indicating that irrigation at late growth stages during November–January helps to maintain canopy growth and increase cassava storage root yield. In contrast, the crops planted in November under rainfed and irrigated conditions were not significantly different for root yield, indicating that irrigation at early growth stages did not increase storage root yield although it did affect canopy development, LAI and light penetration. The study is limited to only one variety with one canopy type, and this research was conducted in only one location. The results in this study could not be extrapolated to other varieties with different canopy types. However, the information obtained in this study is useful to understand canopy development and yield differences under irrigated and rainfed crops planted at different planting dates.

Further studies on different varieties of cassava with different types of canopy architecture, measurement of light intensity in a different layer of canopy and testing in several locations are necessary.

**Author Contributions:** Conceptualization, S.J., P.B., P.T.; C.H.; C.K. and N.V.; methodology, S.J. and N.V.; formal analysis, S.M.; investigation, S.M.; resources, P.B.; data curation, S.M.; writing—original draft preparation, S.M.; writing—review and editing, S.M.; S.J.; P.T.; C.H.; C.K. and T.K.; supervision, S.J.

**Funding:** This research was funded by the Thai Royal Golden Jubilee Ph.D. Program (Grant no. PHD/0031/2559) and the National Science and Technology Development Agency (NSTDA) Thailand.

**Acknowledgments:** This study was supported Assistance in conducting the work was also received from the Plant Breeding Research Center for Sustainable Agriculture, Khon Kaen University, Thailand and Peanut and Jerusalem Artichoke improvement for Functional Food Project. Acknowledgement is extended to the Thailand Research Fund (Project code: IRG5780003) and Faculty of Agriculture, Khon Kaen University for providing financial support for manuscript preparation activities.

**Conflicts of Interest:** The authors declare no conflict of interest

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
