# Peer review of "Seasonal Variations in Canopy Size and Yield of Rayong 9 Cassava Genotype under Rainfed and Irrigated Conditions"

_agronomy, doi:10.3390/agronomy9070362_

Reviewer 1 Report

Intruduction:

Although known, you should describe the reproductive system of Cassava

Materials and Methods: 

What kind of variety is Rayong 9? How was it  selected?Why haven't other control varieties been used…. it would have been useful for the environment  genotype interaction (GxE). The work would have been complete. Results: In the results the significance P≤…(P in italics). Table 2 must be reviewed, in particular the brackets. Are the average values shown in table 2? Add in the caption.  What do the bars in figure 1 indicate? Add in the caption. The same thing in figures 3, 4, 5 and 6

Author Response

Dear Editor/Reviewers

We appreciate the valuable suggestions for further improvement of the manuscript no. Agronomy-514571 entitled "Seasonal variations in canopy size and yield of Rayong 9 cassava genotype under rainfed and irrigated conditions". The changes in the manuscripts are made according to the comments and suggestions of the reviewers. The changes in the manuscript are indicated by "Tag changes". The details of the revision are given below.

The Authors asked Dr. C. C. Holbrook co-author of this manuscript editing English thought out the manuscript.

Best Regards

Sanun Jogloy

Response to Reviewers 1:

Comments and suggestions for authors

Introduction:

1. Although known, you should describe the reproductive system of cassava

More explanation for “Reproductive system of cassava” was added in “Introduction” section. (Page 2, line 44-46)

Materials and Methods:

2. What kind of variety is Rayong 9? How was it selected? Why haven't other control varieties been used…. it would have been useful for the environment 

- More explanation for “Rayong 9 variety” was added in the “Material and Methods” section, subtitle “Plant materials and experimental design” (Page 3, line 85-87)

3. Genotype interaction (G x E). The work would have been complete.

- Thank you for your suggestion.

- The G x E in term of canopy size, light penetration, photosynthesis and yield of 4 cassava genotypes will be reported in a subsequent study. We are preparing that manuscript.

Results:

3. In the results the significance P≤ (P in italics).

- The P≤ (P in italics) in the results or in other sections has been changed.

4. Table 2 must be reviewed, in particular the brackets. Are the average values shown in table 2? Add in the caption. 

- The values in Table 2 are Mean square values and the Values in parentheses are percentages of sum squares. We did add information in the title of table for mean squares and a footnote for values in parentheses (Page 9, line 285)

5. What do the bars in figure 1 indicate? Add in the caption. The same thing in figures 3, 4, 5 and 6

- The bar in figure 1 is rainfall during crop growth in each season and year, but the bars in Figure 3,4, 5 and 6 is standard deviation, that is used to quantify the amount of variation of a set of data values. We did add this information in the title of each figure.

Reviewer 2 Report

Dear Authors,

The manuscript is interesting and makes a valuable contribution to better understanding and conducting research on closing yield gaps in cassava production. The context and significance of the research could be more elaborated, specifically referring to the introduction and conclusion of the paper – what is the relevance of this type of research in view of upgrading farming systems and their socio-economic implications? This would add significantly to the quality of the paper by bringing other perspectives into the picture and thereby addressing aspects of interdisciplinarity, becoming more relevant across the scientific community. The paper is based on extensive literature research.

Detailed comments related to the lines number:

24: As a general remark, the significance of canopy size for cassava production may be outlined

26: RCBD? Needed to spell out

27: 'At final harvest, years were significantly different (P≤0.05) for biomass, shoot dry weight, and harvest index and contributed to large portions of total variations in shoot dry weight (56.8%) and HI (44.5%)' – is this supposed to be “yields”?

31-38: As a general remark to water regimes/supply, it would be helpful to state details on natural water supply (precipitation) during the different seasons.

24-38: Introductory and concluding remark about the implications/significance of the research is missing

44: Only reference to Thailand is made. What about other Asian countries?

80-119: How does Rayong 9 perform in terms of cassava mosaic disease (CMD)? Is it resistant to CMD as a newly developed variety?

78-164: Overall, the part on materials and methods is very well elaborated and clear.

167-195: This is extensive text and rather than a full narrative, it would be more helpful to provide some graphs illustrating the weather conditions during the trials and providing descriptions of the most significant weather patterns during the trials. Also, climate graphs may be used as graphical overlaps with trial results, if appropriate.

225: See comment for line 27

226-456: The narrative of results is very detailed in its description, which is to be rated positively. However, for each section, it would be helpful to summarize the most significant results (3.1, 3.2) possibly in a short paragraph, graph, etc. This would enhance the presentation of results and readability.

459-466: What are the implications of this (the results) for the hypothesis water regime – canopy size and yield?

459-581: the conclusions may add significance by highlighting the implications for improving cassava production systems. This could be stressed more extensively to stress the relevance of the research and required further future research for cassava production systems given the prevailing high yield gaps of the crop (not limited to Thailand). Other factors adding to the prevailing yield gap should also be stated for completeness as water regime and planting date are not the only factors. While recommendations to conduct further research on other varieties and canopy systems are deemed relevant, further studies may take into consideration even other factors of the cropping system as well, i.e. seed use/planting systems (length of cuttings, micropropagation, etc.).

Author Response

Dear Editor/Reviewers

We appreciate the valuable suggestions for further improvement of the manuscript no. Agronomy-514571 entitled "Seasonal variations in canopy size and yield of Rayong 9 cassava genotype under rainfed and irrigated conditions". The changes in the manuscripts are made according to the comments and suggestions of the reviewers. The changes in the manuscript are indicated by "Tag changes". The details of the revision are given below.

The Authors asked Dr. C. C. Holbrook co-author of this manuscript editing English thought out the manuscript.

Best Regards

Sanun Jogloy

Response to Reviewers 2:

Comments and Suggestions for Authors

Dear Authors,

The manuscript is interesting and makes a valuable contribution to better understanding and conducting research on closing yield gaps in cassava production. The context and significance of the research could be more elaborated, specifically referring to the introduction and conclusion of the paper – what is the relevance of this type of research in view of upgrading farming systems and their socio-economic implications? This would add significantly to the quality of the paper by bringing other perspectives into the picture and thereby addressing aspects of interdisciplinary, becoming more relevant across the scientific community. The paper is based on extensive literature research.

 Detailed comments related to the lines number:

1.  L24: As a general remark, the significance of canopy size for cassava production may be outlined

- We added explanation for “the significance of canopy size for cassava production” in “Introduction” section. (Page 2, line 58-60, 60-62 and 69-74)

2. L26: RCBD? Needed to spell out

- This point was mentioned in the “Material and Methods” section, subtitle “Plant materials and experimental design”. (Page 3, line 92)

3. L27: 'At final harvest, years were significantly different (P≤0.05) for biomass, shoot dry weight, and harvest index and contributed to large portions of total variations in shoot dry weight (56.8%) and HI (44.5%)' – is this supposed to be “yields”?

- At final harvest, tuber yields of cassava (economic yield) were not significant difference of the 2 years. The results of this study found that at early growth stage of both planting dates in year 2015 has a warmer temperature and a higher solar radiation than those of the year 2016. At this condition, it promoted  shoot growth rather than tuber root growth [Irikura et al., 1979] then the shoot growth and biomass in 2015 were higher than those in 2016, however at this condition it demoted photo-assimilate partitioning to storage in root tuber then it reduced harvest in 2015.The difference of year for shoot dry weight and HI contributed to the large portion of the total variation for 2 years because of climatic factors, especially temperature  and solar radiation.

- At final harvest, biomass, tuber yields, shoot dry weight and harvest index were significantly different for the 2 planting dates, however, planting date contributed to large portions of total variations in biomass (46.1%) and storage root (60.9%). Cassava planted in November had greater biomass and root yield than did planting in May under both drought and well-irrigated conditions.

4. L31-38: As a general remark to water regimes/supply, it would be helpful to state details on natural water supply (precipitation) during the different seasons.

- This point was presented in the “Discussion section”. (Page 17-18, Line 524-541 and Page 18, 542-551)

5. L24-38: Introductory and concluding remark about the implications/significance of the research is missing

 - More explanation for research missing in “Introduction” section (Page 2, Line 78-80) and implication of the research results on Page 2, line 82-83.

6. L44: Only reference to Thailand is made. What about other Asian countries?

- This point was added in the “Introduction” section. (Page 2, Line 46-48)

7. L80-119: How does Rayong 9 perform in terms of cassava mosaic disease (CMD)? Is it resistant to CMD as a newly developed variety?

- The data of Rayong 9 perform in term of CMD resistance has not been reported yet.

8. L78-164: Overall, the part on materials and methods is very well elaborated and clear.

 - Thank you for your suggestions and comments on this manuscript.

9. L167-195: This is extensive text and rather than a full narrative, it would be more helpful to provide some graphs illustrating the weather conditions during the trials and providing descriptions of the most significant weather patterns during the trials. Also, climate graphs may be used as graphical overlaps with trial results, if appropriate. 

 - Thank you for your suggestions and comments, however we think that the compound graphs showed the real time of crop duration. They were laid out side by side to provide easy comparison between above and below (between two years in the same planting date) or left and right (between 2 planting dates in the same year) compound graphs. We added symbols of each climatic factors in the figure and put the label with bigger letters. 

10. L 225: See comment for line 27

- The responses to reviewer see on line 27

11. L 226-456: The narrative of results is very detailed in its description, which is to be rated positively. However, for each section, it would be helpful to summarize the most significant results (3.1, 3.2) possibly in a short paragraph, graph, etc. This would enhance the presentation of results and readability.

 - Thank you for your suggestions and comments on this manuscript.

- We did summarize the results 3.1 on Page 5 line 175-200 and

- The results of 3.2 were also summarized on Page 5-6 line 201-212.

12. L 459-466: What are the implications of this (the results) for the hypothesis water regime – canopy size and yield?

            The hypothesis of this study is that planting date and water regime are the main causes of differences in canopy size and contribute to yield differences in cassava. (It is indicated in the discussion section on page 16 line 457-458)

            In this study, planting in November resulted in higher biomass and storage root yield than did planting in May under both drought and well-irrigated conditions. (Page 16 line 458-460)

Planting cassava in November has lower temperature and lower relative humidity. In these conditions, low photosynthesis, low canopy growth and high photo-assimilate partitioning to storage in root tuber of cassava were observed. Maximum canopy size was observed 6-7 months after planting.  At that time the temperatures are getting warmer and relative humidity is increasing after getting rain. There is also a high rate of photosynthesis.  In contrast, cassava planting in May is exposed to high temperature and high solar radiation, and photo-assimilate partitioning to shoot and canopy growth reaching the peak of canopy size at 3-4 months after planting (MAP). After four MAP, cool temperature, low% relative humidity and drought stress (case of the rainfed condition) were observed.  Tuber yield from cassava planted in May was lower than planting in November (detail of this information had been presented in  discussion section  on Page 16-17 line 478-489)

Irrigation and rainfall help maintain the canopy size with long leaf size, leaf area life and leaf area duration. These parameters are important for photosynthesis and biomass accumulation. Yield increase in irrigated cassava is proportional to the amount of water applied to the crop, and, if water resources are available, irrigation is an excellent means to increase yield of cassava (detail of this information had been presented in discussion section on Page 18 line 543-549)

13. L459-581: the conclusions may add significance by highlighting the implications for improving cassava production systems. This could be stressed more extensively to stress the relevance of the research and required further future research for cassava production systems given the prevailing high yield gaps of the crop (not limited to Thailand). Other factors adding to the prevailing yield gap should also be stated for completeness as water regime and planting date are not the only factors. While recommendations to conduct further research on other varieties and canopy systems are deemed relevant, further studies may take into consideration even other factors of the cropping system as well, i.e. seed use/planting systems (length of cuttings, micropropagation, etc.).

We did modify this conclusion by deleting some words and adding some suggestions for future research to get more information for improving cassava production in South East Asia countries, especially Thailand. Detail of the conclusion section is revised according to the comments and suggestions (Page 19 line 572-582).

Our results are limited to only one cassava genotype. Cassava varieties may respond differently to water regime and planting date because of genotype by environment interactions. Care must be taken to extrapolate the results to other cassava genotypes and different locations, and more genotypes or plant types should be studied in further investigations to obtain more complete knowledge on this topic. However, our results provide a guideline for further research in order to improve cassava production systems under different environments.

It is well known that cassava has greater yield potential under optimum conditions. However, cassava productivity in many countries is much lower than yield potential, and the large yield gap is due to inappropriate use of cassava varieties, environment conditions and agronomic practices. The new cassava varieties should be of better plant architecture that is well adapted and suitable for different production systems and resistance to important diseases and abiotic stresses such as drought, water logging and nutrient deficiency. The research on growth promoting microorganisms such as Mycorrhiza and phosphate solubilizing microorganisms might help increase resistance to harsh environments. Cassava production is currently dependent on planting of stem cuttings, which will have a drawback if the cuttings are infected with diseases, and, therefore, the research on different propagation methods such as disinfected in vitro propagation and seed propagation as well as the use of different cutting lengths and ages is worth exploring.